# Molecular basis of SAP05-mediated ubiquitin-independent proteasomal degradation of transcription factors

Xiaojie Yan [1,2,7], Xinxin Yuan[1,2,3,7], Jianke Lv[1,4,7], Bing Zhang [1,2,7], Yongle Huang[2], Qianqian Li[1,4], Jinfeng Ma[3], Yanran Li[2], Xiaolu Wang[5], Yao Li [2], Ying Yu [5], Quanyan Liu[3], Tong Liu[6], Wenyi Mi [1,4] ✉ & Cheng Dong [1,2,3,6] ✉

SAP05, a secreted effector by the obligate parasitic bacteria phytoplasma, bridges host SPL and GATA transcription factors (TFs) to the 26 S proteasome subunit RPN10 for ubiquitination-independent degradation. Here, we report the crystal structures of SAP05 in complex with SPL5, GATA18 and RPN10, which provide detailed insights into the protein-protein interactions involving SAP05. SAP05 employs two opposing lobes with an acidic path and a hydrophobic path to contact TFs and RPN10, respectively. Our crystal structures, in conjunction with mutagenesis and degradation assays, reveal that SAP05 targets plant GATAs but not animal GATAs dependent on their direct salt-bridged electrostatic interactions. Additionally, SAP05 hijacks plant RPN10 but not animal RPN10 due to structural steric hindrance and the key hydrophobic interactions. This study provides valuable molecular-level information into the modulation of host proteins to prevent insect-borne diseases.

The ubiquitin-proteasome system (UPS) is the primary pathway for intracellular protein turnover in eukaryotes. It serves as a critical regulator of the proteome by selectively identifying and degrading misfolded, mutated, or excessive proteins[1–3]. Dysregulation of UPS has been implicated in various diseases, such as immune disorders, viral infections, neurodegenerative diseases and cancer[4,5]. In this system, protein substrates are tagged polyubiquitin chains through an enzymatic cascade consisting of ubiquitin-activating enzyme (E1), ubiquitin-conjugating enzyme (E2), and ubiquitin ligase (E3). The substrates are then targeted to the 26 S proteasome for final degradation[6–8].

The 26 S proteasome is a highly conserved multisubunit proteolytic complex composed of a 20 S core particle (CP) and one or both ends capped by 19 S regulatory particles (RP). The 20 S CP is made up of two outer α-rings and two inner β-rings with proteolytic activity. The 19 S RP includes the base and lid subcomplexes: the "base" contains a heterohexameric ATPase ring (RPT1-6) and four non-ATPase subunits (RPN1, 2, 10 and 13), while the "lid" has nine non-ATPase subunits (RPN3, 5–9, 11, 12 and 15)[9–11]. Normally, the RP substrate receptors (RPN1, 10 and 13) capture the ubiquitin chains of substrates, which are then removed by the deubiquitinating enzyme RPN11. Subsequently, the substrate polypeptides are unfolded and translocated by the ATPase ring into the CP for proteolysis[12,13].

Besides the conventional ubiquitin-proteasome pathway, some ubiquitin-independent proteasomal degradation pathways have also been discovered[14,15]. For example, ODC (ornithine decarboxylase) requires modulation via interaction with antizyme (AZ) to be recognized by the proteasome[16,17]. Furthermore, FAT10 (HLA-F locus

[1]The Province and Ministry Co-sponsored Collaborative Innovation Center for Medical Epigenetics, Key Laboratory of Immune Microenvironment and Disease (Ministry of Education), School of Basic Medical Sciences, Tianjin Medical University, Tianjin 300070, China. [2]Department of Biochemistry and Molecular Biology, Tianjin Medical University, Tianjin 300070, China. [3]Department of Hepatobiliary Surgery, Tianjin Medical University General Hospital, Tianjin 300052, China. [4]Tianjin Institute of Immunology, Department of Immunology, School of Basic Medical Sciences, Tianjin Medical University, Tianjin 300070, China. [5]Department of Pharmacology, Tianjin Key Laboratory of Inflammatory Biology, Center for Cardiovascular Diseases, Tianjin Medical University, Tianjin 300070, China. [6]Department of Cardiology, Tianjin Institute of Cardiology, Second Hospital of Tianjin Medical University, Tianjin 300211, China. [7]These authors contributed equally: Xiaojie Yan, Xinxin Yuan, Jianke Lv, Bing Zhang. ✉e-mail: wenyi.mi@tmu.edu.cn; dongcheng@tmu.edu.cn

adjacent transcript 10) is a ubiquitin-like modifier added to a substrate in a process reminiscent of the ubiquitin pathway in that it also requires E1/E2/E3 ligases and the presence of lysine residues on the substrate targeted for degradation, and the NUB1L (NEDD8 ultimate buster-1 long) accelerates FAT10-mediated degradation. Whereas FAT10 and NUB1L bind to the vWA (von Willebrand factor A) domain of RPN10, and are degraded along with their substrates[18,19]. Additionally, Huang and colleagues recently reported that plant pathogenic phytoplasmas secrete an effector, SAP05, which concurrently mediates the degradation of the plant SPL and GATA developmental transcription factors (TFs) in a ubiquitin-independent manner by hijacking the plant host 26 S proteasome subunit RPN10. This infection converts plants into "zombie plants" with spectacular vegetative organ proliferation and juvenilization, serving as habitats for the phytoplasma pathogens and their insect vectors[20–24].

In this ubiquitin-independent pathway, SAP05 bridges plant SPL and GATA TFs to the N-terminal vWA domain of RPN10 rather than the C-terminal UIM (ubiquitin-interacting motifs) domain for degradation[20]. Remarkably, although animals also encode GATA TFs and RPN10 proteins, and the zinc finger domains of GATAs, or vWA domains of RPN10s are highly conserved from plants to animals, SAP05 does not bind animal GATAs and RPN10s[20]. However, the detailed information on SAP05-mediated recognition of SPL and GATA TFs as well as RPN10 remains largely elusive.

In this study, we present the first crystal structures of SAP05 in complex with SPL, GATA or RPN10. Our findings show that SAP05 acts as a natural "PROTAC" (proteolysis targeting chimera) degrader using two opposing surfaces to link the TFs directly to the RPN10 for degradation. Our high-resolution structures, combined with mutagenesis and degradation assays, provide a detailed molecular mechanism explaining how SAP05 selectively targets plant host TFs and 26 S proteasome, while avoiding animal TFs and RPN10. Furthermore, our findings offer alternative amino acids of TFs or RPN10 that could be engineered to resist parasite effector activity.

## Results

### Overall crystal structures of SAP05 bound to transcription factors and AtRPN10

SAP05 is an effector from a plant pathogenic phytoplasma that mediates the degradation of host transcription factors, including SPL and GATA family proteins, by hijacking the 26 S plant ubiquitin receptor RPN10 independently of substrate ubiquitination[20]. To explore the SAP05-mediated binding model, the *Arabidopsis thaliana* (At) zinc-finger (ZnF) domains of GATA18 and SPL5, and the SAP05 from onion yellows phytoplasma (OY-M; Candidatus Phytoplasma asteris) without the secretory signal peptide (SP) were purified individually from *Escherichia coli* (Fig. 1a). The vWA domain of AtRPN10 and SAP05 were co-purified together, as the individual AtRPN10 tends to aggregate in the absence of SAP05 assistance. SAP05 was found to form a stable binary complex with SPL5 or GATA18, as well as the corresponding ternary complex with AtRPN10 during the size exclusion chromatography (Fig. 1b and Supplementary Fig. 1a). The binding affinities of SAP05 for GATA18 and SPL5 were quantified using isothermal titration calorimetry (ITC) assays, which showed robust binding with a $K_D$ value of 0.14 μM and 0.42 μM, respectively (Fig. 1c, d).

To gain insight into the molecular interactions of SAP05-mediated complex, we successfully determined the binary crystal structures of SAP05-GATA18, SAP05-SPL5 and SAP05-AtRPN10, albeit the concurrent ternary complex was not available. Data collection and model refinement statistics are summarized in Table 1. It is noteworthy that, to the best of our knowledge, SAP05 adopts a protein fold since without a structure similarity in the DALI server[25], since the search did not yield any close matched to the SAP05 structure. Specifically, SAP05 folds into a tight globular structure consisting of a parallel five-stranded β-sheet (β1-β5). This central β-sheet is sandwiched between

two α-helices flanking one side (α1-α2) and one helix flanking the other side (α3) (Fig. 1e).

SAP05 can bridge two distinct host TF families to AtRPN10, based on our structures, the ZnF domains of GATA and SPL families both bound to SAP05 in a similar manner, although they do not share very similar structural characteristics (Fig. 1e, f). In the SAP05-GATA18 binary complex, the ZnF of GATA18 has a type IV zinc finger motif containing a single zinc $CysX_2CysX_{18}CysX_2Cys$ zinc finger[26], in which the zinc ion is chelated by four cysteine residues (Cys154, Cys157, Cys176 and Cys179) located on the α-helix and the extended loop (Fig. 1e). This zinc finger motif is evolutionarily conserved from plants to animals[27,28].

In the SAP05-SPL5 binary complex, the ZnF of SPL5 possesses two zinc-binding subdomains, of which N-terminal subdomain contains an extended loop followed by two short helices to coordinate one $Cys_3His$-type zinc finger motif, whereas the C-terminal subdomain employs a three-stranded antiparallel β-sheet and a short helix to coordinate a $Cys_2HisCys$-type zinc finger motif (Fig. 1f). It should be noted that, unlike other types of zinc-binding domains, these two zinc-binding sequence motifs are not interleaved, thereby forming a unique plant-specific TF called SQUAMOSA promoter binding protein (SBP)[29].

In the SAP05-AtRPN10 crystal structure, one asymmetric unit contains two SAP05-AtRPN10 binary complexes with almost identical architecture. Interestingly, the N-terminal loop (L1) of SAP05 becomes a β-strand by packing against its counterpart, forming an antiparallel β-sheet in the crystallographic packing. (Supplementary Fig. 1b). The vWA domain of AtRPN10 harbors a conserved topology that resembles those of other orthologs across species[30]. Briefly, the vWA domain consists of a central six-stranded β-sheet with five parallel strands and one antiparallel strand (β3) flanked by three α-helices on each side (Fig. 1g). Appropriately, the α1 and α2 helices of vWA domain, which are not directly involved in the interactions with other subunits of the 19 S regulatory particle[31], pack against SAP05 (Supplementary Fig. 2a, b).

Although SAP05 carries a single domain, it utilizes two opposing surfaces (hereafter referred to as Lobe1 and Lobe2) to interact simultaneously with TFs and AtRPN10, forming a dumbbell-shaped structure, in which TFs and AtRPN10 have no direct surface contacts. This notion is further supported by our ITC assays, as the binary complex AtRPN10-SAP05 and individual SAP05 exhibited comparable binding affinities against GATA18 (0.10 μM vs 0.14 μM) or SPL5 (0.37 μM vs 0.42 μM) (Supplementary Fig. 1c, d). Therefore, SAP05 shows some similarities with "PROTAC" that brings together individual TFs and AtRPN10 by means of two distinct lobes (Fig. 1h and Supplementary Fig. 1e).

### Molecular interactions of SAP05 with TFs and AtRPN10

Upon inspection of the complex structures, we found that the Lobe1 of SAP05 creates an acidic surface that interacts with the complementary basic surface of TFs (Fig. 2a, b, e and f). For instance, in the SAP05-GATA18 structure, the guanidine group of Arg188 from GATA18 forms a bidentate salt bridge with the carboxyl group of Asp106 from SAP05. Simultaneously, the Lys185 of GATA18 is salt-bridged to the negatively charged Asp106 and Glu107 of SAP05. Furthermore, Arg182 of GATA18 forms another bidentate salt bridge with Asp108 of SAP05. Apart from electrostatic interactions, the α-helix of GATA18 is further stabilized through hydrophobic interaction between Ile181 of GATA18 and Ile84-Trp85 of SAP05. At the edge of the α-helix of GATA18, the side-chain carboxamide of Asn177 is coordinated by two hydrogen bonds with the carboxyl group of Asp66 and the backbone carbonyl of Phe81 from SAP05 (Fig. 2c, d). Thus, the basic α-helix of GATA18 is tightly bound to the acidic path of SAP05 through a combination of electrostatic interactions, hydrophobic interactions and hydrogen bonds.

In the SAP05-SPL5 structure, both subdomains of SPL5 participate in the interactions with SAP05. Particularly, for the N-terminal subdomain's contacts, a hydrophobic core is formed at the interface

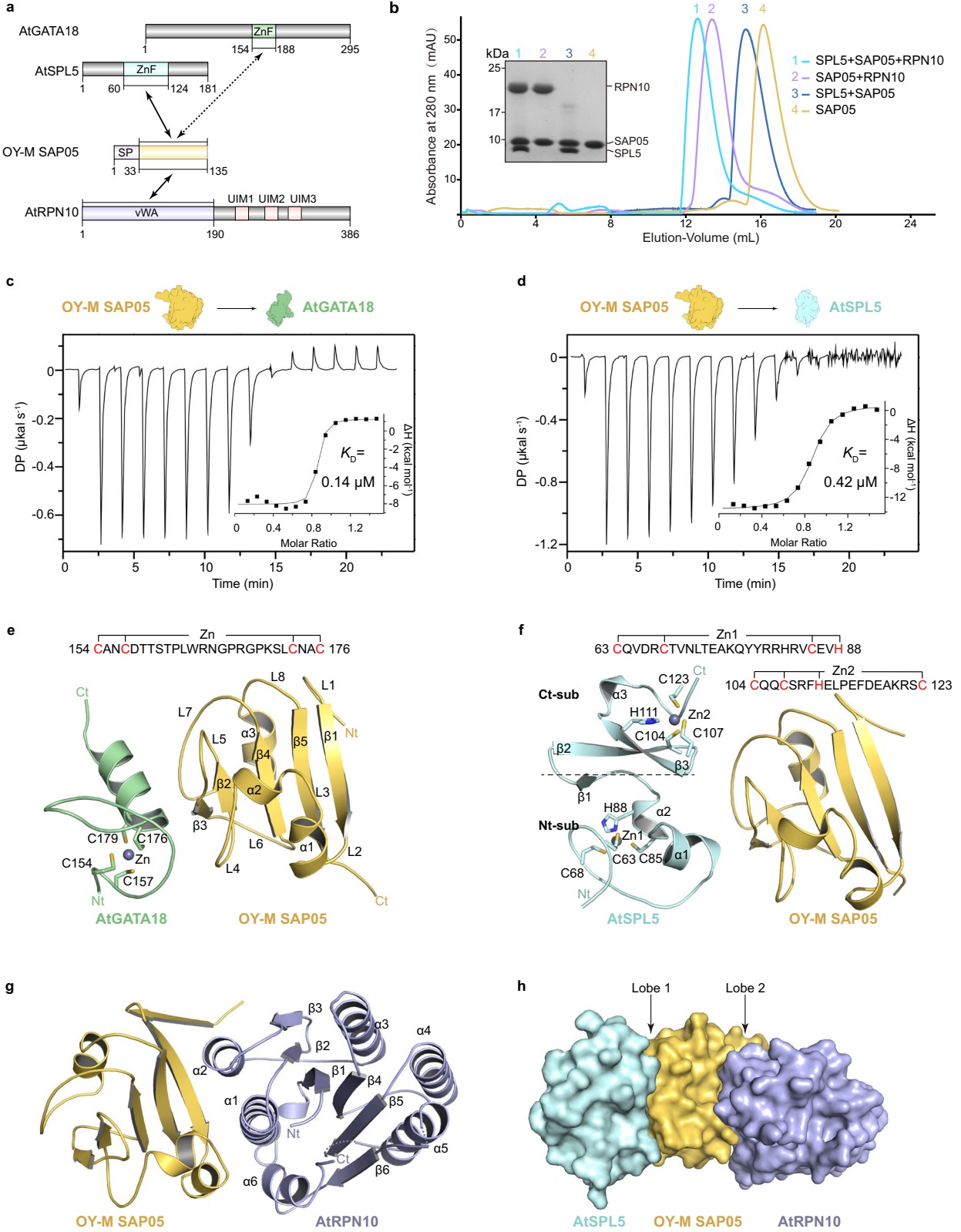

between the α1 of the N-terminal subdomain donated by Tyr78 and Tyr79 and the β3 of SAP05 donated by Ile84 and Trp85. Besides, the Lys90 from α2 of the N-terminal subdomain forms a bidentate salt bridge with the carboxyl group of Glu107 from SAP05. Likewise, for the C-terminal subdomain's contacts, a bidentate salt bridge is created by the Arg121 and the Asp106. This interface is further strengthened by

three direct hydrogen bonds contributed by Gln105 and Ser108 of the C-terminal subdomain with Arg104 and Asn77 of SAP05 (Fig. 2g, h).

In addition to SPL5, SAP05 can also target plant SPL13, another member of the SPL family that plays an important role in the juvenile-to-adult vegetative transition and the vegetative-to-reproductive transition[32]. By X-ray crystallography, we also obtained the complex

**Fig. 1 | Overall structures of SAP05 bound to TFs and RPN10. a** Domain architecture of *Arabidopsis thaliana* (At) GATA18, AtSPL5, onion yellows phytoplasma (OY-M) SAP05 and AtRPN10. ZnF, zinc finger; SP, signal peptide; vWA, von Willebrand factor A; UIM, ubiquitin-interacting motif. **b** Superdex 75 Increase 10/300 gel-filtration chromatography profiles of SAP05 alone and in complex with SPL5, RPN10 and both. Left panel, SDS-PAGE gel of the peak fractions, stained with Coomassie blue. Source data are provided as a Source Data file. Representative images, *n* = 3. **c, d** ITC measurements of binding affinities ($K_D$) of OY-M SAP05 to AtGATA18 and AtSPL5 ZnF domains, respectively. **e** Ribbon diagram of the crystal

structure of OY-M SAP05-AtGATA18 complex. Zinc ion is shown as a sphere. Four zinc-coordinating cysteine residues of AtGATA18 are numbered and shown in schematic form. **f** Crystal structure of OY-M SAP05-AtSPL5 complex. AtSPL5 coordinates two non-interleaved Zn1 and Zn2 sites and is separated into an N-terminal subdomain (Nt-sub) and a C-terminal subdomain (Ct-sub). **g** Crystal structure of OY-M SAP05-AtRPN10 complex. **h** Surface diagram of the complex SPL5-SAP05-RPN10 by the superposition of SAP05 from the single complex SPL-SAP05 and SAP05-RPN10. The contact surfaces of SPL-SAP05 and SAP05-RPN10 are referred to as Lobe1 and Lobe2, respectively.

## Table 1 | Data collection and refinement statistics

| PDB accession number | GATA18-SAP05 8J48 | SPL5-SAP05 8J49 | SAP05-RPN10 8J4A | SPL13-SAP05 8J4B |
|---|---|---|---|---|
| Data collection | | | | |
| Space group | P 2₁ | P 1 | P 2₁ | P 2₁2₁2 |
| Cell dimensions | | | | |
| a, b, c (Å) | 35.84, 73.41, 58.69 | 30.16, 45.09, 52.95 | 54.05, 40.59, 130.33 | 90.92, 63.35, 68.56 |
| α, β, γ (°) | 90.00, 107.22, 90.00 | 92.43, 101.86, 108.62 | 90.00, 93.01, 90.00 | 90.00, 90.00, 90.00 |
| Resolution (Å) | 36.70-1.94 (2.01-1.94)* | 42.44-1.66 (1.75-1.66) | 48.95-1.97 (2.04-1.97) | 36.93-2.00 (2.07-2.00) |
| $R_{sym}$ or $R_{merge}$ | 0.133 (1.006) | 0.034 (0.250) | 0.066 (0.468) | 0.109 (0.627) |
| $I / \sigma I$ | 10.79 (1.65) | 11.70 (2.30) | 11.87 (2.85) | 13.74 (3.21) |
| Completeness (%) | 97.2 (81.6) | 83.5 (32.5) | 94.6 (98.1) | 99.5 (96.7) |
| Redundancy | 6.1 (4.1) | 3.4 (2.9) | 2.8 (2.6) | 9.1 (7.2) |
| Refinement | | | | |
| Resolution (Å) | 36.70-1.94 (2.01-1.94) | 42.44-2.10 (2.18-2.10) | 42.66-1.97 (362.04-1.97) | 34.28-2.00 (2.07-2.00) |
| No. reflections | 20846 (1743) | 14114 (1389) | 38214 (3983) | 27304 (2608) |
| $R_{work}$ / $R_{free}$ | 0.1961/0.2371 | 0.1748/0.2113 | 0.1925/0.2526 | 0.1753/0.2132 |
| No. atoms | | | | |
| Protein | 2267 | 2191 | 4596 | 2747 |
| Ligand/ion | 2 | 2 | 0 | 4 |
| Water | 187 | 180 | 477 | 284 |
| B-factors | | | | |
| Protein | 28.2 | 24.4 | 14.5 | 28.5 |
| Ligand/ion | 31.7 | 29.1 | 0 | 38.5 |
| Water | 33.4 | 29.7 | 22.0 | 34.9 |
| R.m.s. deviations | | | | |
| Bond lengths (Å) | 0.009 | 0.008 | 0.008 | 0.009 |
| Bond angles (°) | 1.01 | 0.96 | 1.12 | 0.94 |

*Values in parentheses are for highest-resolution shell.

structure of SAP05 with SPL13, structural analysis indicated that it shares a conserved interaction pattern observed in the SAP05-SPL5 structure (Supplementary Fig. 3a, b), highlighting the evolutionary conservation of SPL family recognition by SAP05.

In contrast to the acidic Lobe1, the Lobe2 possesses a strong hydrophobic path for AtRPN10 binding (Fig. 2i, j), where a cluster of hydrophobic residues including Leu35, Leu73 and Leu69 on the α1-2 helices of AtRPN10 plus Phe49, Phe58, Ile47, Phe61 and Tyr127 on the Lobe2 lining the central interface. While at the edge of α1 helix of vWA domain, the side chains of Gln27 and Glu31 are accommodated by hydrogen bonds with the backbones of Thr48 and Ser50 of SAP05, respectively (Fig. 2k, l). Of note, a number of water molecules permeated the Lobe2 and vWA interface, with multiple molecules mediating contacts between SAP05 and AtRPN10 (Supplementary Fig. 3c). As a result, this binary complex buries a total of 833Å² of solvent accessible surface area.

In summary, SAP05 bridges the TFs mainly through electrostatic interactions with a hydrophobic core, whereas the SAP05-AtRPN10 interface comprises predominantly extensive hydrophobic contacts.

Additionally, several hydrogen bonds and water-mediated interactions also contribute to the formation of the ternary complex.

## Mutagenesis studies of SAP05-bridged interactions with TFs and AtRPN10

To determine the precise roles of the key residues in the formation of the ternary complex, we generated a series of single point mutants and evaluated the effect on binding. To this end, we performed ITC experiments to examine the binding affinities of mutant SAP05 toward wild-type TFs, and vice versa. Unsurprisingly, these mutants resulted in varying degrees of reduction in binding activity, ranging from mild to complete loss of binding ability. For example, in the SAP05-SPL5 interface, substitution of SAP05 Glu107 or Asp106 with alanine, which would disrupt the salt-bridged interaction, resulted in 15-fold decreased or abolished binding affinity for SPL5. Moreover, the alanine substitution of SAP05 Trp85, impairing hydrophobic interaction, caused a complete loss of SPL5 binding (Fig. 3a). In parallel, alanine replacement of the corresponding electrostatic or hydrophobic interacting Arg121, Tyr78 or Tyr79 in SPL5 abrogated its ability to bind

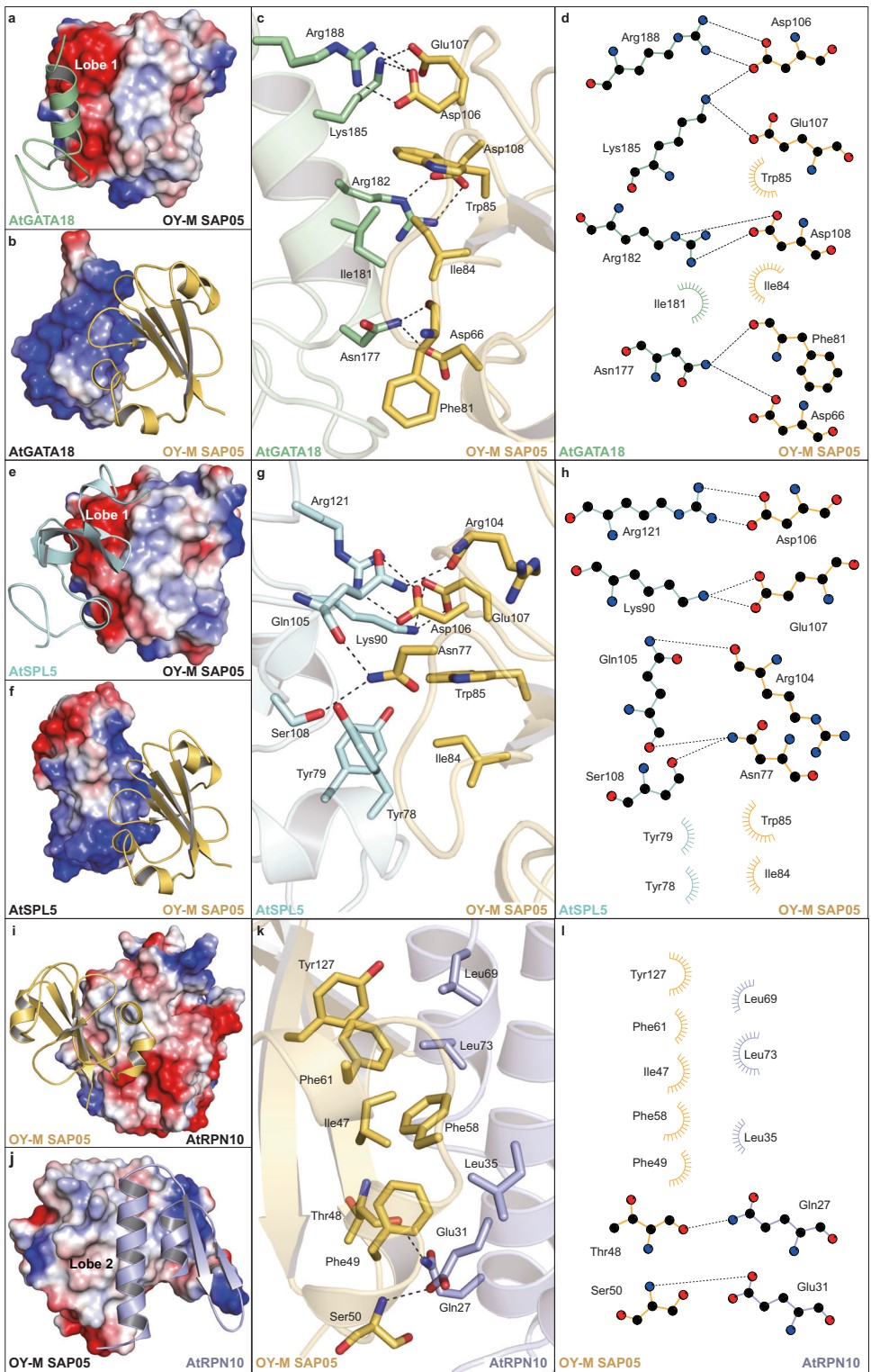

**Fig. 2 | The interaction details of SAP05-mediated contacts with TFs and RPN10.**
**a**, **b** The electrostatic potential surface of the individual OY-M SAP05 and AtGATA18 (red, negative; blue, positive). **c** Close-up view of the interactions of OY-M SAP05 with AtGATA18. Residues of AtGATA18 and OY-M SAP05 that are involved in the interactions are shown as palegreen and yelloworange sticks, respectively. The hydrogen bonds are shown as black dashed lines. **d** Ligplot diagram illustrating the contacts between OY-M SAP05 and AtGATA18. The hydrophobic contacts are indicated as semicircles with radiating spokes. **e**, **f** The electrostatic potential surface of the individual OY-M SAP05 and AtSPL5. **g** Close-up view of the interactions of OY-M SAP05 with AtSPL5. Residues of AtSPL5 and OY-M SAP05 that are involved in the interactions are shown as palecyan and yelloworange sticks, respectively. **h** Ligplot diagram illustrating the contacts between OY-M SAP05 and AtSPL5. **i**, **j** The electrostatic potential surface of the individual OY-M SAP05 and AtRPN10. **k** Close-up view of the interactions of SAP05 with AtRPN10. Residues of SAP05 and AtRPN10 that are involved in the interactions are shown as yelloworange and lightbule sticks, respectively. **l** Ligplot diagram illustrating the contacts between OY-M SAP05 and AtRPN10.

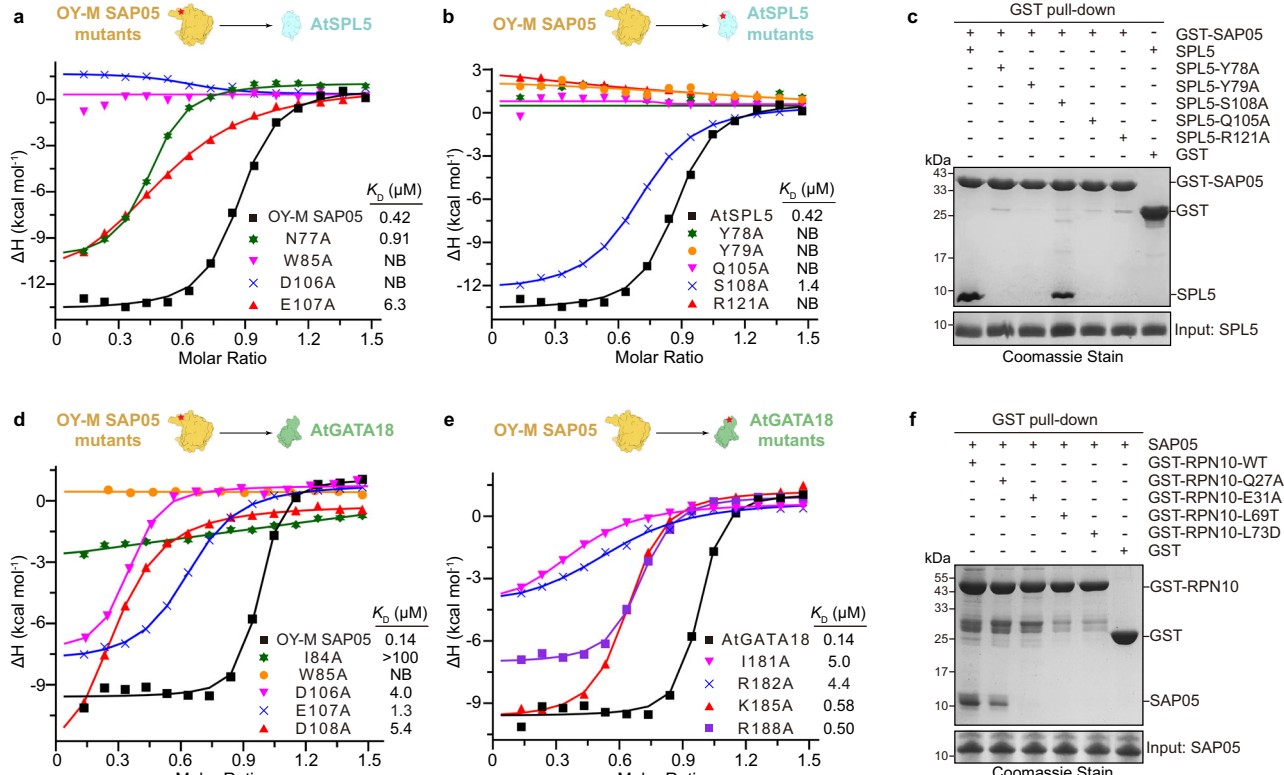

**Fig. 3 | Mutation analysis of the interactions SAP05 with TFs and RPN10. a** ITC fitting curves of wild-type and mutant OY-M SAP05 titrated to wild-type AtSPL5. NB, no apparent binding under our experimental conditions. **b** ITC fitting curves of wild-type OY-M SAP05 titrated to wild-type and mutant AtSPL5. **c** GST pull-down assay using GST-tagged OY-M SAP05 to pull down wild-type and mutant AtSPL5. Source data are provided as a Source Data file. Representative images, $n = 3$. **d** ITC fitting curves of wild-type and mutant OY-M SAP05 titrated to wild-type AtGATA18. **e** ITC fitting curves of wild-type OY-M SAP05 titrated to wild-type and mutant AtGATA18. **f** GST pull-down assay using GST-tagged wild-type and mutant AtRPN10 to pull down co-purified wild-type OY-M SAP05. Source data are provided as a Source data file. Representative images, $n = 3$.

SAP05. On the other hand, the Q105A and S108A mutants, in which the hydrogen bonds were ablated, exhibited defects in SAP05 binding (Fig. 3b). These above observations were further supported by our GST pull-down assays (Fig. 3c and Supplementary Fig. 4a). At the same time, size exclusion chromatography analysis indicated that the SPL5 mutants, such as R121A and Q105A, abolished the coelution of the ternary complex AtRPN10-SAP05-SPL5 (Supplementary Fig. 4b, c), suggesting that electrostatic, hydrophobic interactions and hydrogen bonding are important for the interactions between SAP05 and SPL5, which is consistent with our structural analyses.

In the SAP05-GATA18 interface, three pairs of salt-bridge interactions stabilize this binary complex, whereas a single alanine mutation of the negatively charged residue (Asp106, Glu107 or Asp108) in SAP05 or the positively charged residue (Arg182, Lys185 or Arg188) in GATA18 diminished the binding affinity by 3- to 40-fold (Fig. 3d, e). In addition, the hydrophobic interaction defective mutants, such as W85A and I84A in SAP05 or I181A in GATA18 mutant, led to a significant reduced interaction (Fig. 3d, e). Together, these results established that these key residues of SAP05 (such as Asp106 and Glu107, especially Trp85) play a critical role in mediating both SPL5 and GATA18 binding.

To test the importance of the residues involved in SAP05-AtRPN10 interaction, we carried out GST pull-down assay using the wild-type and mutant AtRPN10 tagged with GST to pull down the co-purified SAP05. The results showed that the mutant E31A, L69T or L73D of AtRPN10 was unable to pull down SAP05 (Fig. 3f), suggesting the prominent role of Glu31-mediated hydrogen bonding, and Leu69-, Leu73-mediated hydrophobic interactions in the recognition of SAP05.

To further verify whether SAP05 mutants are intact and remain correctly folded in our assay, we carried out a GST pull-down assay by using GST-RPN10 to pull down the SAP05 mutants that are defective in TFs binding in our ITC assay. The results indicated that the wild-type SAP05 and the TFs-binding-deficient mutations could apparently be pulled down by GST-RPN10, but two negative control samples SAP05-F58D (a mutant defective in RPN10 binding) and GFP could not be pulled down by GST-RPN10 (Supplementary Fig. 4d), indicating that these SAP05 mutants retain RPN10-binding activity albeit loss of TF-binding activity. In addition, gel-filtration chromatography assay showed that all SAP05 mutants exhibit a uniform non-oligomerization peak identical to the wild type (Supplementary Fig. 4e), suggesting that SAP05 mutants are stable and correctly folded.

## The key residues for SAP05-mediated degradation of SPL and GATA TFs

To further corroborate whether these interacting residues are required for SAP05-mediated TFs degradation, we carried out an in vitro degradation assay using purified human 26 S proteasome as previously reported[20]. In this assay, despite the fact that the purified 26 S proteasomes incorporate some endogenous human RPN10, the introduction of additional purified vWA can successfully compete with endogenous RPN10 for binding to the 26 S proteasomes (Supplementary Fig. 5a). The results showed that wild-type SAP05 is capable of bridging SPL5 for ubiquitin-independent degradation in the presence of AtRPN10 vWA domain with the purified human 26 S proteasomes, while neither addition of proteasome inhibitor MG132 nor in the absence of AtRPN10 vWA domain or human 26 S proteasome was able to promote SPL5 degradation (Fig. 4a and Supplementary Fig. 5b–d).

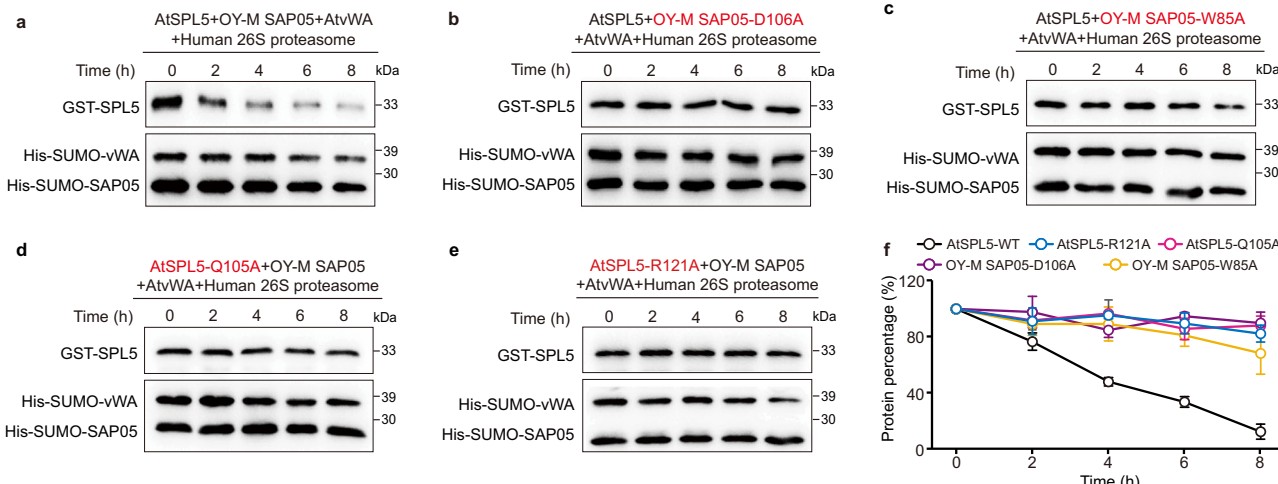

**Fig. 4 | Mutagenesis analysis of SAP05-mediated degradation of TFs. a** Western blot analysis of OY-M SAP05-mediated degradation of AtSPL5 using purified human 26 S proteasomes in the presence of wild-type AtSPL5, OY-M SAP05 and AtvWA (AtRPN10 vWA domain). **b**–**e** Western blot analysis of the degradation of AtSPL5 in the presence of OY-M SAP05 mutants (D106A or W85A) or AtSPL5 mutants (Q105A or R121A). Source data are provided as a Source Data file. Representative images, $n = 3$. **f** Quantification of the percentage of retained GST-SPL5 in the degradation assay, corresponding to (**a**–**e**) (Mean ± S.E.M.; $n = 3$ independent experiments).

As expected, the mutants of SAP05, such as D106A or W85A, which lost SPL5 binding ability, were unable to promote SPL5 degradation (Fig. 4b, c and f). Conversely, the loss-of-binding mutants of SPL5, such as Q105A or R121A, were not degraded by SAP05 (Fig. 4d, e and f). Likewise, the binding-defective mutants W85A and I84A of SAP05 were less efficient in inducing SAP05-mediated degradation of GATA18 compared to wild-type protein (Supplementary Fig. 5e–h). In addition, the mutants that compromised the SAP05-AtRPN10 interface, such as E31A, L69T and L73D in AtRPN10, were severely retarded in SAP05-mediated degradation of SPL5 (Supplementary Fig. 5i–l). Taken together, these data further support our findings that the key residues involved in SAP05-SPL5, SAP05-GATA18 and SAP05-AtRPN10 interactions play a vital role in the SAP05-mediated degradation of TFs through 26 S proteasome.

**Engineering SAP05 homolog to alter its recognition specificity**
SAP05 homologs are present in divergent phytoplasmas, and some SAP05 homologs have evolved to differentially interact with SPL and GATA TFs. For instance, OY-M SAP05 (this study) interacts with both SPLs and GATAs. However, WBDLa SAP05 (one copy from witches' broom disease of lime) recognizes only SPLs and the other copy, WBDLb SAP05, recognizes only GATAs[20]. To address why WBDLa SAP05 is unable to target GATAs, we analyzed the residues required for GATA18 binding through sequence alignment, and found that the majority of residues are strictly conserved between OY-M SAP05 and WBDLa SAP05, except for Ile84 in OY-M SAP05, which is replaced by Met86 in WBDLa SAP05 (Fig. 5a). So we mutated Met86 of WBDLa SAP05 to Ile and examined the interaction by ITC. However, the single-point mutant M86I was insufficient to bind GATA18 (Fig. 5b). Next, we aligned the Alphafold-predicted WBDLa SAP05 structure with OY-M SAP05 in the SAP05-GATA18 complex structure, and observed that the larger Met86 likely causes a steric hindrance impeding GATA18 binding (Fig. 5c). In addition to Met86, we also found that Thr68 in OY-M SAP05 is replaced by the longer side chain Lys70 in WBDLa SAP05, which gives rise to another steric clash, although this residue is not directly involved in GATA18 binding (Fig. 5d). Indeed, the single K70T mutant was not sufficient to bind GATA18, whereas the double mutant M86I·K70T exhibited a strong binding capacity to GATA18 ($K_D = 2.7 \mu M$) but still retained SPL5 binding ($K_D = 2.6 \mu M$) (Fig. 5b and Supplementary Fig. 6a). Meanwhile, our degradation assay revealed that, in contrast to the wild-type WBDLa SAP05, its double mutant can

bridge the GATA18 for degradation in the 26 S proteasome (Fig. 5e–g). Correspondingly, the double mutant T68K and I84M in OY-M SAP05 lost its intrinsic GATA18 binding activity, and failed to induce the GATA18 degradation (Supplementary Fig. 6b, c). Of note, AT SAP05 (Candidatus Phytoplasma mali), like WBDLa SAP05, which does not target GATAs, also carries a Met and Lys at the equivalent position (Fig. 5a), resulting in a steric clash with GATAs. Therefore, the SAP05 homologs have evolved to selectively recognize GATA TFs through structural steric hindrance, and the counterparts of Thr68 and Ile84 in OY-M SAP05 play an important role in the selection criteria of GATA TFs.

**Mechanism by which Phytoplasma SAP05 targets host rather than its insect vector GATA TFs**
Phytoplasma secretes effector SAP05 to target plant GATAs and SPLs TFs without attacking their insect vector GATAs (SPLs are absent in animals)[20,33]. However, the mechanism behind this phenomenon remains elusive. Through sequence alignment and structural analysis of AtGATA and leafhopper vector MqGATA (Macrosteles quad-rilineatus), we found that the three positively charged residues Arg182, Lys185 and Arg188 in AtGATA, which form multiple salt bridges with SAP05, are not conserved in MqGATA (Fig. 5h, i). Although single mutants (R182A, K185A or R188A) in AtGATA still retain the ability to bind SAP05 (Fig. 3e), its triple mutant completely lost of SAP05 binding (Supplementary Fig. 6d). As anticipated, swapping of these three counterparts within MqGATA (Tyr, Met and Asn) with those positively charged residues present in the AtGATA displayed a strong binding affinity with SAP05 ($K_D = 0.90 \mu M$) (Fig. 5j). As expected, this triple-mutant rather than wild-type MqGATA could be degraded by SAP05 (Fig. 5k–m). Consistently, mutating this three positively charged residues in AtGATA was unable to be targeted by SAP05 for degradation (Supplementary Fig. 6e). Therefore, the phytoplasma SAP05 effector takes advantage of the salt-bridged interactions to distinguish host from its insect vector GATA TFs.

**Mechanism of SAP05 specifically hijacking host RPN10**
Phytoplasma SAP05 effector mediates plant TFs for degradation by hijacking the AtRPN10. However, SAP05 does not target insect vector RPN10, despite RPN10 being highly conserved among eukaryotes (Fig. 6a)[20]. Intriguingly, swapping AtRPN10 two conserved residues to insect vector MqRPN10 residues (38GA39 > HS) resulted in loss of

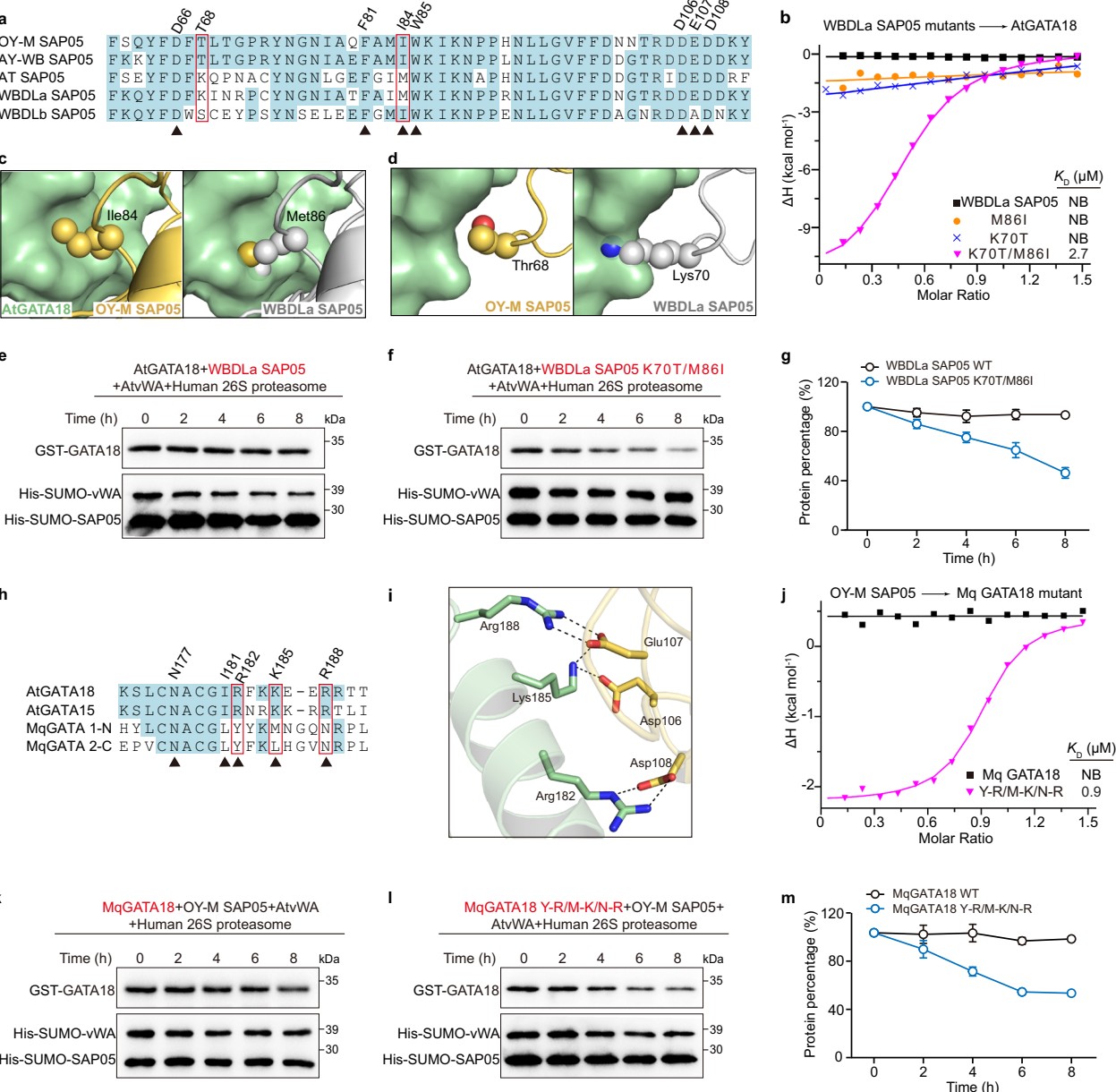

**Fig. 5 | The key residues of SAP05 and GATA18 for their specific recognition and degradation. a** Sequence alignment of SAP05 homologs in divergent phytoplasmas. Identical residues are marked by lightblue background. The residues that directly interact with GATA18 are numbered and denoted by black triangles. The residues that cause steric hindrance are indicated by red frames. **b** ITC measurements of binding affinities of wild-type and mutant WBDLa SAP05 to AtGATA18. NB, no apparent binding under our experimental conditions. **c** Structural comparison of the effect of the substitution of Ile84 in OY-M SAP05 by Met86 in WBDLa SAP05 on steric hindrance. The corresponding residues are shown as spheres. **d** Structural comparison of the effect of the substitutions of Thr68 in OY-M SAP05 by Lys70 in WBDLa SAP05 on steric hindrance. **e, f** Western blot analysis of the degradation of AtGATA18 using purified human 26 S proteasomes in the presence of wild-type or double mutant WBDLa SAP05. Source data are provided as a Source Data file.

Representative images, $n = 3$. **g** Quantification of the percentage of retained GST-GATA18 in the degradation assay, corresponding to (**e, f**) (Mean ± S.E.M.; $n = 3$ independent experiments). **h** Sequence alignment of GATA domains from *A. thaliana* and *M. quadrilineatus*. Residues directly involved in SAP05 interactions in the OY-M SAP05-AtGATA18 structure are numbered and denoted by black triangles. **i** Three positively charged residues of AtGATA18 rather than MqGATA form salt bridge interactions. **j** ITC measurement of binding affinity of OY-M SAP05 to wild-type and mutant MqGATA18 (Y-R, M-K, N-R). **k, l** Western blot analysis of OY-M SAP05-mediated degradation of wild-type or mutant MqGATA18 (Y-R, M-K, N-R). Source data are provided as a Source Data file. Representative images, $n = 3$. **m** Quantification of the percentage of retained GST-GATA18 in the degradation assay, corresponding to (**k, l**) (Mean ± S.E.M.; $n = 2$ or 3 independent experiments).

SAP05 binding in the yeast two-hybrid assay, but the underlying mechanism remains largely unclear. In light of our SAP05-AtRPN10 structure, we found that residues 38GA39, located on the α1 helix of AtRPN10, do not directly participate in the interaction with SAP05, albeit they are in close proximity to the SAP05 surface. However, the substituted His and Ser residues, especially the bulky His, would induce steric hindrance with the SAP05 surface (Fig. 6b). Indeed, like the 38GA39 > HS mutant, the single-site mutant G38H of AtRPN10 resulted in the loss of the AtSAP05 binding, but the A39S mutant of AtRPN10 retained the AtSAP05 binding activity (Fig. 6d). Closer examination of the crystal complex structure, we discovered that, the Gly70 located on the α2 helix of AtRPN10 also packs closely against the

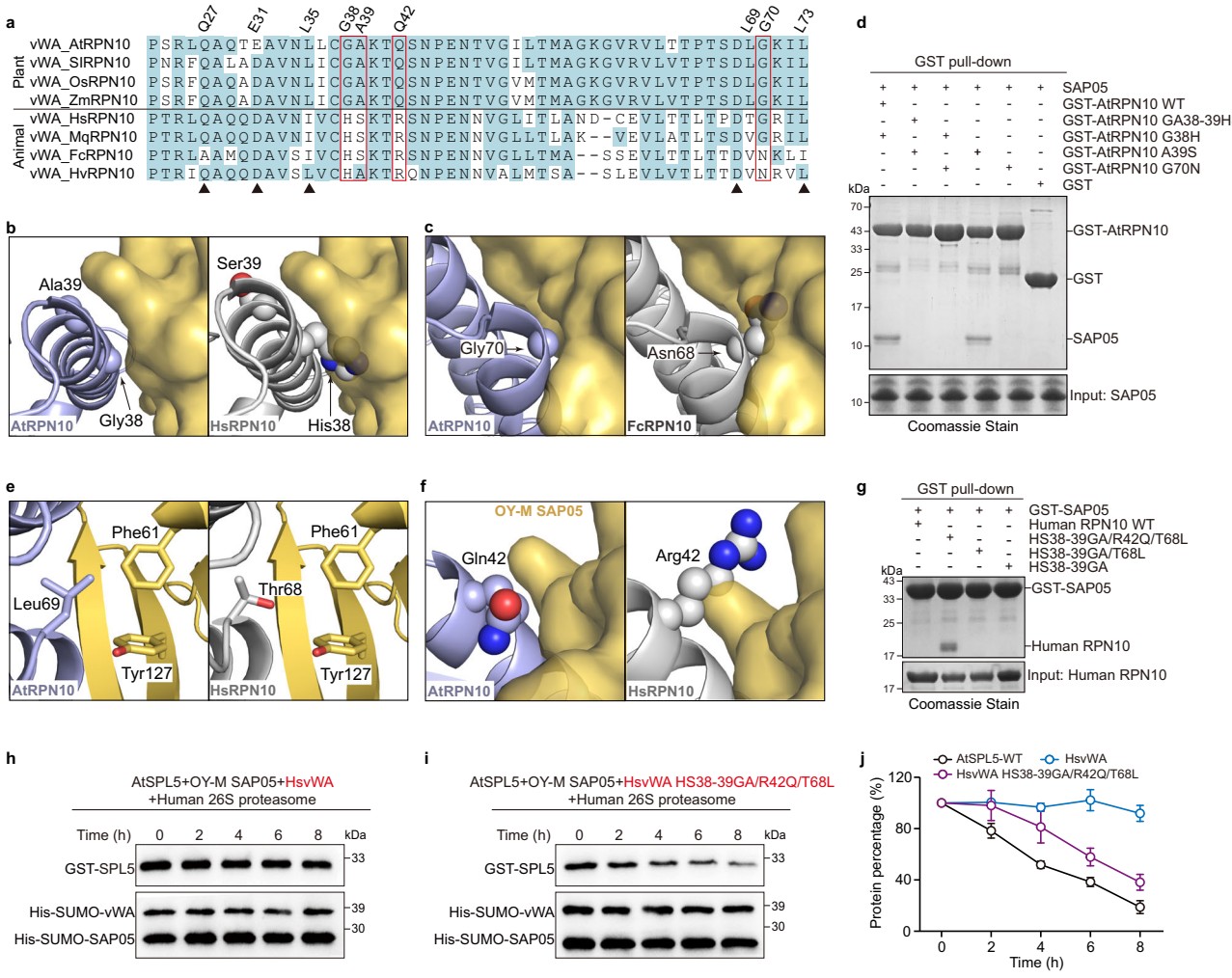

**Fig. 6 | The key residues of RPN10 for SAP05 binding. a** Sequence alignment of RPN10 homologs between plants and animals. AtRPN10, *Arabidopsis thaliana* RPN10 (Uniprot ID: P55034); SlRPN10, *Solanum lycopersicum* RPN10 (Uniprot ID: A0A3Q7F6N7); OsRPN10, *Oryza sativa* RPN10 (Uniprot ID: O82143); ZmRPN10, *Zea mays* RPN10 (Uniprot ID: B6TK61); HsRPN10, *Homo sapiens* RPN10 (Uniprot ID: Q5VWC4); MqRPN10, *Macrosteles quadrilineatus* RPN10 (GenBank: XP_054291051.1); FcRPN10, *Folsomia candida* (Uniprot ID: A0A226E266); HvRPN10, *Hydra vulgaris* (Uniprot ID: T2MF29). The residues involved in SAP05 binding are marked with black triangles. **b** Structural analysis of steric hindrance caused by His38 of human RPN10 (PDB: 6MSE). **c** Structural analysis of steric hindrance caused by Asn68 of FcRPN10 (corresponding to Gly70 in AtRPN10). The structure of FcRPN10 is predicted by Alphafold2, and superimposed with AtRPN10 in the SAP05-AtRPN10 complex. **d** GST pull-down assay using wild-type and mutant AtRPN10 to pull down OY-M SAP05. Source data are provided as a Source Data file. Representative images, *n* = 3. **e** Structural analysis of the absence of hydrophobic interaction caused by Thr68 in human RPN10 instead of Leu69 in AtRPN10. **f** Structural analysis of steric hindrance caused by Arg42 of human RPN10. **g** GST pull-down assay using OY-M SAP05 to pull down wild-type and mutant human RPN10. **h**, **i** Western blot analysis of OY-M SAP05-mediated degradation of SPL5 in the presence of wild-type and mutant human vWA domain (HS38-39GA, R42Q, T68L) instead of AtRPN10 vWA domain in purified human 26 S proteasomes. Source data are provided as a Source Data file. Representative images, *n* = 3. **j** Quantification of the percentage of retained GST-SPL5 in the degradation assay, corresponding to **h**, **i** and Fig. 4a (Mean ± S.E.M.; *n* = 3 independent experiments).

SAP05 surface, similar to Gly38 (Fig. 6c). As anticipated, substitution of Gly70 in the AtRPN10 to the Asn present in other animal species (e.g. *Springtail* and *Hydra*) can prevent recognition and degradation by SAP05 as well (Fig. 6a, d and Supplementary Fig. 7a).

Given that animal RPN10 harbors a conserved His-Ser context prevented recognition by SAP05, we hypothesized that whether SAP05 could target animal RPN10 by residue-swapping mutagenesis. We first generated a 38HS39 > GA mutant on human RPN10, but this substitution is not sufficient to bind SAP05 (Fig. 6g). Comparison of vWA domains of human and plant homologs indicated that most of the residues involved in SAP05-AtRPN10 are conserved, except for the hydrophobic Leu69, which is replaced by Thr68 in human (Fig. 6a). Our GST pull-down assay revealed that the conserved Leu69 of AtRPN10 is required for SAP05 binding (Fig. 3f), as it contributes strong hydrophobic interaction (Fig. 6e). Nevertheless, simultaneous

mutation of 38HS39 > GA and T68L could not rescue the SAP05 binding (Fig. 6g). Structural comparison of the AtRPN10 and human RPN10 showed that the highly conserved Gln42 in AtRPN10 is altered by a longer side-chain Arg42 in animal RPN10, which would cause extra steric clash in addition to His38 (Fig. 6f). Mutating 38HS39 > GA in combination with T68L and R42Q within human RPN10 successfully binds to SAP05 by means of GST pull-down and gel filtration assays (Fig. 6g and Supplementary Fig. 7b). In support of this observation, this multiple-site mutant of human RPN10, but not wild-type human RPN10, can be directly targeted by SAP05 for SPL5 degradation in the human 26 S proteasome without AtRPN10 (Fig. 6h–j).

In summary, our complex structures, together with binding and degradation assays, suggest that phytoplasma SAP05 effector has evolved to take advantage of a unique structural features to selectively target host TFs to RPN10 for degradation. Some SAP05 homologs

target only host SPL, but not GATAs due to structural steric hindrance. In addition, the lack of direct salt-bridged electrostatic interactions shields SAP05 from recognizing its insect vector GATAs. SAP05 cannot hijack animal RPN10 due to structural steric hindrance and the absence of the key hydrophobic interactions.

## Discussion

In eukaryotic cells, the UPS is responsible for the degradation of more than 80% of intracellular proteins[34]. This system utilizes ubiquitin-tagging of targeted substrates as a signal to regulate this programmed process[12,22]. Notably, the phytoplasma SAP05 can target plant SPL and GATA TFs for degradation by binding directly to the proteasome substrate receptor RPN10, without the need for substrate ubiquitination[20]. Herein, we elucidate the molecular mechanisms underlying the recognition of SAP05-bridged TFs (SPL5 and GATA18) to RPN10 and the subsequent degradation of TFs in the 26 S proteasome. SAP05 functions like a "PROTAC" by bridging individual TFs and AtRPN10 via two opposing lobes, forming compact ternary complexes through bilateral electrostatic interactions, hydrophobic interactions and hydrogen bonds. During the preparation of this manuscript, a preprint presenting the crystal structures of SAP05-SPL5 and SAP05-RPN10 was available on bioRxiv[35]. Additionally, the crystal structure of SAP05-RPN10 was reported in another preprint[36].

Of note, SAP05 engages TFs for degradation but avoids degradation itself. An explanation is that during the initiation of TFs translocation into the proteasome CP chamber for degradation, SAP05 is expected to uncouple with TFs but remain with RPN10 possibly due to the latter has a stronger bond with SAP05. Alternatively, in the processive translocation phase, the TFs undergo unfolding by the heterohexameric ATPase ring, and the unfolded TFs lose their ability to bind to SAP05. Consequently, SAP05 can be released from TFs to spare them from degradation. Nevertheless, a small amount of SAP05 and/or vWA components are still co-transported with TFs into the CP channel for degradation, since a slight degradation of SAP05 and vWA was observed in our western blot assay.

SAP05 is an "intrusive protein" that disturbs the vegetative phase transition in plants, leading to a slower growth and cessation of reproduction.by promoting the degradation of SPL and GATA TFs[20]. Our study uncovered that SAP05 utilizes its acidic surface to tightly bind to the basic surface of the ZnF domain of GATA18 or SPL5 TFs. We also identified several single-residue mutations in SPL5 or GATA18 that can completely abolish the SAP05-TFs binding, providing a potential strategy for rational engineering of plant TFs to block SAP05 activity and enhance host resistance to this phloem-inhabiting, insect-vectored bacterial pathogen.

Different SAP05 homolog subclades exhibit binding and degradation to both TFs or only SPLs and GATAs, which may be attributed to the difference in evolutionary time of the host-microbe interactions or diverging survival requirements of the microbes[20,37]. We provide a structural explanation for the selective binding of WBDLa SAP05 to SPLs rather than GATAs, where the Met86 and Lys70 residues of WBDLa SAP05 induce dual structural steric hindrance during its interaction with AtGATA18. In contrast to WBDLa SAP05, WBDLb SAP05 can bind to GATAs, but not to SPLs[20]. We generated some mutants by swapping the single or multiple residues of WBDLb SAP05 with those present in the other SAP05 homologs did not rescue the SPL5 binding ability (Supplementary Fig. 8), suggesting that a more complex mechanism may be involved.

Previous studies have shown that AtRPN10 GA > HS mutant is resistant to SAP05 activity during phytoplasma infection[20]. Our structural analysis indicated that the individual bulky His substitution in AtRPN10 induces steric hindrance in its binding with SAP05, we also identified additional potential sites, such as Gly70, Glu31, Leu69 and Leu73 of AtRPN10 that could be engineered to block SAP05 activity,

paving the way for further engineering of plants to protect against insect-borne diseases.

## Methods

### Protein expression and purification

The genes encoding OY-M SAP05 (residues 33-135), AtSPL5 ZnF domain (residues 60-124), AtSPL13 ZnF domain (residues 98-162), AtGATA18 ZnF domain (residues 1-100), AtRPN10 vWA domain (residues 1-193) and human RPN10 vWA domain (residues 2-193) were separately cloned into pET28-MKH8SUMO (Addgene plasmid #79526) vector with an N-terminal 8×His-SUMO tag followed by a TEV cleavage site. These recombinant plasmids were transformed into *Escherichia coli* strain BL21 (DE3) cells with Kanamycin selection. Cells were cultured in Luria-Bertani liquid medium at 37 °C until the $OD_{600}$ reached ~0.6, then protein expression was induced at 18 °C overnight with a final concentration of 0.2 mM isopropyl-β-D-1-thiolgalactopyr-anoside (IPTG). After collection by centrifugation, the well-cultured cells expressing OY-M SAP05 and vWA domain of AtRPN10 were mixed at a ratio of 1: 2 and co-lysed by sonication on ice in lysis buffer containing 20 mM Tris-HCl pH 7.5, 400 mM NaCl, 2 mM β-mercaptoethanol, other protein-expressing cells were lysed individually. Cell debris was removed by centrifugation at 14,000 rpm for 40 min at 4 °C. The clarified supernatant was collected and loaded onto a Ni-NTA column, non-specific binding protein was washed with 25 mM imidazole in lysis buffer and target protein was eluted with 300 mM imidazole in lysis buffer. The SUMO tag was removed by addition of TEV protease at a mass ratio of 1: 30 in 20 mM Tris-HCl pH 7.5, 300 mM NaCl dialysis buffer at 4 °C overnight and then the samples were reloaded onto the Ni-NTA column to obtain untagged protein. Further purification was performed by gel filtration chromatography using a Superdex 75 Increase 10/300 size exclusion column (GE Healthcare) pre-equilibrated with gel-filtration buffer containing 20 mM Tris-HCl pH 7.5, 150 mM NaCl and 1 mM dithiothreitol (DTT). For the purification of the SPL5-SAP05, SPL13-SAP05 and GATA18-SAP05 complexes, SAP05 and its binding partner were mixed at a molar ratio of 1:2, and then incubated on ice for one hour. SAP05-AtRPN10 complex was obtained by co-purification. The complex can be separated by gel filtration and confirmed by SDS-PAGE. Fractions containing the target protein were concentrated to approximately 10 mg/mL and frozen at −80 °C for storage. Mutants were generated by the QuikChange site-directed mutagenesis strategy using the wild-type plasmid as a template and verified by DNA sequencing. Purification of the protein variants was consistent with that of the wild-type.

### Protein crystallization

Crystallization trials were performed at 18 °C using commercially available kits and were carried out by the sitting-drop vapor diffusion method with mixing 1 μL reservoir solution and 1 μL protein sample. Diffraction-quality crystals were obtained under the following conditions:

GATA18-SAP05: 0.2 M Lithium acetate dehydrate and 20% (*w/v*) polyethylene glycol 3350;

SPL5-SAP05: 0.2 M Sodium fluoride and 20% (*w/v*) polyethylene glycol 3350;

RPN10-SAP05: 0.2 M Lithium acetate dehydrate, 18% (*w/v*) polyethylene glycol 3350 and 0.1 M guanine hydrochloride;

SPL13-SAP05: 0.2 M Ammonium acetate, 0.1 M HEPES pH 7.5 and 25% (*w/v*) polyethylene glycol 3350.

Crystals were mounted directly in the respective well solutions supplemented with 25% (*v/v*) glycerol and flash frozen in liquid nitrogen for data collection.

### Data collection and structure determination

Diffraction data were collected on the beamline BL02U or BL18U at Shanghai Synchrotron Radiation Facility (SSRF) and processed with

XDS[38]. The complex structures of GATA18-SAP05, SPL13-SAP05 and SPL5-SAP05 were solved by the single wavelength anomalous diffraction (SAD) method using the SAD data generated with zinc containing proteins GATA18, SPL13 and SPL5. Experimental phasing was generated using the AutoSol and AutoBuild programs in the PHENIX suite[39,40]. The auto-built models from the phasing programs were manually rebuilt using Coot[41] and refined by PHENIX[39]. The complex structure of SAP05- AtRPN10 was solved by molecular replacement with Phaser using SAP05 structure as a search template. All figures were calculated and prepared using the PyMOL program (https://www.pymol.org).

## Isothermal titration calorimetry

ITC assays were carried out using a MicroCal PEAQ-ITC instrument (Malvern Panalytical) at 16 °C. All proteins were prepared in ITC buffer which containing 20 mM Tris-HCl pH 7.5, and 150 mM NaCl. For each experiment, 15 injections of 1.5 μL protein (except for the first injection using 0.5 μL protein) with concentration ranged from 660 μM - 700 μM were titrated into sample cell containing 40 μM to 50 μM protein, with a 90 s interval time between the injections. The results were analyzed using the MicroCal PEAQ-ITC Analysis Software version 1.30. The data were fitted with a one-site model. Each experiment was repeated independently at least three times with similar results.

## GST pull-down assay

Genes encoding AtRPN10, OY-M SAP05 and their mutants were cloned separately into pGEX vector with an N-terminal GST tag. The sequence encoding OY-M SAP05 was cloned into pET28-MHL vector with an N-terminal 6×His tag. Fusion proteins were expressed in *E. coli* BL21 cells. All the proteins except GST-AtRPN10/mutants and His-SAP05 were purified by standard Ni-NTA or GST affinity chromatography, and further purified by gel-filtration using a Superdex 200 Increase 10/300 GL column (GE healthcare).

Approximately 200 μL of GST-AtRPN10 and His-SAP05-expressing cells were co-lysed by sonication in binding buffer containing 20 mM Tris-HCl, 400 mM NaCl, and then the supernatant clarified by concentration was incubated with 10–20 μL GST beads for 1 h at 4 °C on a rotating wheel. For the other GST pull-down assay, approximately 200 μg of corresponding GST-fused protein or GST alone were incubated with 10–20 μL GST beads followed by incubation with equal amounts of the corresponding target protein for 1 h at 4 °C. After three washes with binding buffer, the samples were eluted with the same buffer with addition of 25 mM glutathione. The eluates were analyzed by SDS-PAGE gel followed by Coomassie blue staining.

## Plasmid construction of 26 S proteasome

His-TEV-Biotin-His (HTBH) DNA fragment was synthesized by Bgi Genomics Company and cloned into pLVX according to the following amino acid sequence: HHHHHHDYDIPTTASENLYFQSELKTAALAQHDEAAGKAGEGEIPAPLAGTVSKILVKEGDTVKAGQTVLVLEAMKMETEINAPTDGKVEKVLVKERDAVQGGQGLIKIGHHHHHH. Human Rpn11 (hRpn11) was amplified by PCR using cDNA as a template with the following primers: forward, 5′CCCTCGAGCTATGGACAGACTTCTTAGACTT3′, reverse, 5′CGGAATTCGATTTAAATACGACAGTATCCAAC3′. PCR products were inserted into pLVX-HTBH by *Xho* I and *Eco*R I restriction enzyme digestion. Finally, a plasmid encoding the human 26 S proteasome subunit RPN11, tagged with a biotin epitope at the C-terminus, was produced.

## Cell culture

HEK293T cells were maintained according to the manufacturer's recommendations. Cells were passaged every 2 days and maintained in a humidified cell culture incubator at 37 °C with 5% CO₂. A HEK293T cell line was co-transfected with recombinant lentiviral vector (pLVX-hRPN11-HTBH) and packaging vectors (psPAX2 and pMD2.G) by LipoInsect™ Transfection Reagent. Lentivirus was produced and released into the medium between 48 h after transfection, harvested and passed through a 0.45 μm filter. HEK293T cells were seeded in 6-well plates and infected with the medium containing the lentivirus in the presence of 8 μg/mL polybrene for 48 h. Stable cell lines expressing hRpn11-HTBH were subsequently selected with 2 μg/mL puromycin.

## Affinity purification of human 26 S proteasomes

An affinity purification strategy was carried out as reported[42,43]. The biotinylated derivatives tagged RPN11 were used to rapidly isolate the human 26 S proteasome complex for subsequent degradation assay. Briefly, a stable HEK293T cell line expressing His-Biotin-Tev-His (HBTH)-tagged RPN11 was generated and maintained. The collected cells were lysed in buffer A (100 mM sodium chloride, 50 mM sodium phosphate, 10% glycerol, 5 mM ATP, 1 mM DTT, 5 mM MgCl₂, 1 × protease inhibitor (Roche), 1 × phosphatase inhibitor, and 0.5% NP-0.4 (pH 7.5)) for 30 min at 4 °C, and then the lysates were centrifuged at 13,000 rpm for 15 min. The protein concentration of supernatant was measured by the Bradford method using G250 kit (Beyotime, P0006). Approximately 2 mg of supernatant extract was incubated with 10 μL streptavidin beads (bead concentration: 10 mg/mL, the binding capacity for free biotin: >1100 pmol/mg) overnight at 4 °C. The streptavidin beads were washed five times with buffer A, followed by one wash with TEB buffer (50 mM Tris-HCl pH 7.5 and 10% glycerol). Finally, the beads of per tube were resuspended with 40 μL TEB buffer and stored at −80 °C.

## Degradation assay in human 26 S proteasomes

The purified human 26 S proteasomes (containing hRPN11 beads) were used immediately after thawing. 1.25 μg GST-SPL5 wild type or its mutants (or GST-GATA18, GST-MqGATA18 or its mutants) and 10 μg SAP05-vWA wild-type complex (or only SAP05) or its mutants were added to 8 μL of purified 26 S proteasomes containing beads in 100 μL reaction buffer (50 mM Tris-HCl pH 7.5, 50 mM NaCl, 10 mM MgCl₂, 10% glycerol, 2 mM DTT and 5 mM ATP)[20], and then incubated at 28 °C. The concentration information of these proteins was provided in the supplementary materials. At the incubation time for 0, 2, 4, 6 and 8 h, 20 μL aliquots was collected for each reaction. The 26 S proteasome activity was inactivated with 100 μM MG132. Collected samples were added with SDS-PAGE loading buffer, boiled immediately and stored at −20 °C until used for western blotting analysis. The efficiency of degradation or turnover rates of protein were detected by the amounts of substrates at different incubation times. The amounts of substrates were determined by gray scale of protein bands from three independent experiments unless otherwise stated, using ImageJ software (National Institute of Health). Values are presented as the mean ± S.E.M. (Standard Error of the Mean).

## Western immunoblotting

Protein degradation was measured by western blotting. Samples were separated by SDS−PAGE and transferred to PVDF membrane (Immobilon-P, Millipore), blocked with 5% skim milk in TBST. Membranes were incubated with primary antibody overnight at 4 °C, washed three times, and incubated with secondary antibody for 1 h at room temperature. Blots were detected by an ECL system. His (ZSGB-Bio, TA-02, 1:1000), GST (Beyotime, AF5063, 1:2000) or RPN10 (Proteintech, 14899-1-AP, 1:1000) antibodies were used to detect His-, GST-fusion proteins or RPN10, respectively.

## vWA competition assay

Different molar ratios of vWA: purified human 26 S proteasomes (0:1, 2:1, 4:1), were added to the 50 μL reaction buffer (500 mM Tris-HCl pH 7.5, 500 mM NaCl, 100 mM MgCl₂, 2 mM DTT and 5 mM ATP), and then incubated at 28 °C for 2 h. Collected supernatant and beads were

added with SDS-PAGE loading buffer, boiled immediately and stored at −20 °C until used for western blotting analysis.

## Reporting summary

Further information on research design is available in the Nature Portfolio Reporting Summary linked to this article.

## Data availability

The atomic coordinates for the reported structures have been deposited in the Protein Data Bank (PDB) under accession codes 8J48 (GATA18-SAP05), 8J49 (SPL5-SAP05), 8J4A (SAP05-RPN10), 8J4B (SPL13-SAP05). All study data are included in the article and/or SI Appendix. Source data are provided with this paper.

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

## Acknowledgements

We thank the staff at beamlines BL02U and BL18U of Shanghai Synchrotron Radiation Facility for assistance in X-ray data collection. This work was supported by Shandong Province Special Fund "Frontier Technology and Free Exploration" from Pilot National Laboratory for Marine Science and Technology (Qingdao) (No. 8-01), National Natural Science Foundation of China grants 32271265 (to C.D.), 32071193 (to C.D.), 82321001 (to C.D.), 82173000 (to W.M.) and 82103176 (to X.W.), National Youth Top-Notch Talent Support Program in China, Tianjin Municipal Science and Technology Commission grant 22JCZDJC00440 (to C.D.), Research Foundation of Tianjin Municipal Education Commission grants 2021ZD036 (to C.D.) and 2022KJ191 (to B.Z.), and Core Facility of Research Center of Basic Medical Sciences at Tianjin Medical University.

## Author contributions

C.D. and W.M. conceptualized the project and designed experiments. X.J.Y. and X.X.Y. cloned the constructs and performed protein expression, purification, crystallization, GST pull-down and ITC assays with the help from Y.H., J.M., Y.R.L. and Y.L. X.J.Y. determined the crystal structures. J.L., Q.Q.L. and X.W. carried out the degradation experiments assays. C.D., B.Z., Y.Y., Q.Y.L., T.L. and W.M. analyzed the data. C.D. wrote the manuscript with critical inputs from all authors.

## Competing interests

The authors declare no competing interests.
