## [Peer Review File · Nature Communications]

Molecular basis of SAP05-mediated ubiquitin-independent proteasomal degradation of transcription factorsREVIEWER COMMENTS

Reviewer #1 (Remarks to the Author):

A recent study has demonstrated that SAP05 effector of bacterial phytoplasma mediates the degradation of plant transcription factors (TFs), such as SPL5 and GATA18, by the 26S proteasome in a ubiquitin-independent manner. Here, Yan et al solved the crystal structures of SAP05-SPL5, SAP05-GATA18, or SAP05-RPN10 subunit of Arabidopsis 26S proteasome. Their structural analysis, combined with mutational, binding, and protein degradation assays, reveal that the SAP05 form two separate interfaces to interact with TFs and RPN10, providing a molecular understanding of how SAP05 mediates ubiquitin-independent proteasomal degradation of plant TFs. In addition, their structural modeling analyses explain why SAP05 targets RPN10 and host TFs in a species-specific manner. Overall, the crystal structures are of high resolution. The central conclusion is largely supported by the structural and biochemical evidence. I only have a few minor concerns on the structural and biochemical data.

1. The authors claim the structure of SAP05 to be a new fold with a β 1-strand dissociated from the β 2-strand. The lack of intramolecular interaction between the β 1-strand and the rest of the protein is very unusual. Is this due to a crystallization artifact? The authors need to present all the protein complexes, if more than one, in one asymmetric unit as a supplementary figure.
2. For reliable comparison of binding affinities, the ITC results for all the repeats need to be averaged with the standard deviation being provided, either in the figures or as a supplementary table. In addition, the binding stoichiometry (N value) needs to be provided. In Fig. 5i, the ITC binding assays for SAP05 with WT MqGATA18 needs to be presented for comparison.
3. In Table 1, the I/σ for the highest-resolution shell of 8J48 and 8J49 entries is much higher than 2. On what considerations were the resolution cutoffs selected?
4. There is a mis-incorporated number '1.5' in Supplementary Fig. 6a.

5. In discussion, the authors mentioned that they failed to identify why WBDLb SAP05 does not bind to SPLs. Yet no experimental data were presented. The authors may want to elaborate more details on their work in this regard.

Reviewer #2 (Remarks to the Author):

In this manuscript, Yan et al. investigate the molecular interactions of SAP05 with SPL and GATA transcription factors and RPN10. The work follows up on a previous study from another lab that describes the mechanism of how phytoplasma SAP05 effector mediates the degradation of SPLs and GATAs by recruiting the 26S proteasome receptor RPN10. Huang et al found that SAP05 does not interact with animal GATA and insect/human RPN10 homologs and showed that purified human 26S proteasomes can degrade plant SPLs in the presence of SAP05 and Arabidopsis RPN10. Yan et al built on this knowledge to generate crystal structures and identify residues that may play a role in the specificity of SAP05 binding to the plant proteins.

Except for providing details of molecular interactions, the study does not reveal new mechanistic insights. For example, the work does not reveal how the 26S proteasome is degrading SPL and GATA targets.

The study shows additional insights into which amino acids play a role in the interactions of SAP05 and RPN10. However, the work is incomplete. Firstly, the study lacks data that show that the SAP05 mutants are intact and remain correctly folded. To address this concern, it is necessary to demonstrate that the mutant proteins retain other activities and that their overall stability remains unaffected in the assays.

Data described in lanes 215 to 220 of the results misrepresent the data shown. This section claims that SAP05 mutants are studied, but the figure shows data for SPL5 mutants.

The majority of the 26S proteasome experiments included in the paper don't show evidence of degradation, and the few that may do, there is only marginal differences between the lanes, and moreover, the protein levels in the other lanes in the same experiments appear to go down as well. The 26S proteasome experiments include limited information on

turnover rates and efficiency of degradation in relation to the concentrations of substrates provided.

Reviewer #3 (Remarks to the Author):

The present work approaches the mechanism of SAP05-mediated degradation by the proteasome of plant transcription factors SPL5 and GATA18 by means of crystal structure and structure/function studies.

SAP05 is a very new player in research literature, with only three articles in PUBMED, starting in 2021. The main reference on SAP05 is an excellent work published in Cell (Huang W, MacLean AM, Sugio A, Maqbool A, Busscher M, Cho ST, Kamoun S, Kuo CH, Immink RGH, Hogenhout SA. Parasitic modulation of host development by ubiquitin-independent protein degradation. Cell. 2021 Sep 30;184(20):5201-5214.e12. doi: 10.1016/j.cell.2021.08.029. Epub 2021 Sep 17. PMID: 34536345; PMCID: PMC8525514.). Hogenhout group's work describes a very relevant and unprecedented mechanism of degradation of transcription factor by SAP05, hijacking one of the main receptors of the 26S proteasome, Rpn10, for directing substrates to the proteasome in an ubiquitination-independent fashion. Without going deeper on, it is clear that the mechanism uncovered by Hogenhout's paper is not only important in parasite biology, but also in proteasome, protein degradation and proteostasis fields, since well-detailed ubiquitin-independent protein degradation processes by the 26S proteasome are scarce in the literature.

Said that, the current paper becomes highly relevant since it complements and characterizes the mechanism proposed by Huang and collaborators. Furthermore, it describes with great detail the interactions between complex partners and the basis for the specificity of the mechanism. The experimental design and the methodology are outstanding and appropriate for the work. The results are remarkable, with several crystal structures solved, including subsequent interface definition, mutagenesis and functional assays. Furthermore, in general terms, it is correctly written and references are fine.

My suggestion is that the paper should be accepted with very minor revisions. However, there are aspects that should be addressed:

- It is mentioned that, SAP05 "acts", "behaves" or "functions" as a "PROTAC" that brings together individual TFs and AtRPN10 by means of two distinct lobes (Lines 82, 147 and 363).

This analogy generates problems. Protacs are usually small-molecule based chimeras chemically synthesized that recognize specific motifs in targets, and they recruit E3-ligases to induce ubiquitination. SAP05 would act more as an adaptor with some analogies with the mechanism of recruitment and degradation of ornithine decarboxylase (ODC), or with the interaction FAT10 - Rpn10-vWA (even though the involved surfaces are not identical). The "PROTAC" concept could be maintained in the text, but in different terms. For example, by changing "behaves like a PROTAC" by "shows some similarities with proximity-based proteolytic chimeras" or similar sentences. Moreover, it should be highlighted in the text that it is not explained how SAP05 avoids degradation. In the case of small Protacs it is evident that they are not proteins so they are not proteolyzed by the 26S, but SAP05 is a protein that binds Rpn10. Why it is not degraded?

Furthermore, ODC and FAT10 should be mentioned somewhere in the paper.

- Line 108: "SPL5-AtRpn10" should be "SAP05-AtRpn10"

- Line 256: "In eukaryotic cells, the UPS is responsible for the degradation of more than 80% of intracellular proteins" should have a reference.

- Proteasome purification and degradation assays: the way they are designed, human 26S proteasome are purified and vWA-Rpn10 from *Arabidopsis thaliana* is used to reconstitute the degradation (by ternary interactions with SAP05 and substrates). This is acceptable, and apparently it works, however it is not mentioned if purified 26S proteasomes are occupied by human Rpn10. It is important to check if human Rpn10 is stoichiometric, sub-stoichiometric or absent, in proteasomes purified for the assay. In methods section, the washing conditions are defined as: "streptavidin beads were washed 5 times with buffer A" (containing 100 mM NaCl). It is hard to know if endogenous Rpn10 will be retained or dissociated from the 26S, therefore, this information should be specified.

- Figure 4: it is too crowded and repetitive. I would recommend to limit it to 3-4 panels and the rest in the supplement.

- All Figures with ITC graphs: "moler ratio" should be "molar ratio"

Response to Reviewer #1

Comments:

A recent study has demonstrated that SAP05 effector of bacterial phytoplasma mediates the degradation of plant transcription factors (TFs), such as SPL5 and GATA18, by the 26S proteasome in a ubiquitin-independent manner. Here, Yan et al solved the crystal structures of SAP05-SPL5, SAP05-GATA18, or SAP05-RPN10 subunit of Arabidopsis 26S proteasome. Their structural analysis, combined with mutational, binding, and protein degradation assays, reveal that the SAP05 form two separate interfaces to interact with TFs and RPN10, providing a molecular understanding of how SAP05 mediates ubiquitin-independent proteasomal degradation of plant TFs. In addition, their structural modeling analyses explain why SAP05 targets RPN10 and host TFs in a species-specific manner. Overall, the crystal structures are of high resolution. The central conclusion is largely supported by the structural and biochemical evidence. I only have a few minor concerns on the structural and biochemical data.

Response:

Thank you very much for your positive comments and constructive suggestions on our manuscript. We have provided the detailed data and relevant explanation that you requested to strengthen our manuscript. All the concerns are addressed as follows.

1. The authors claim the structure of SAP05 to be a new fold with a β 1-strand dissociated from the β 2-strand. The lack of intramolecular interaction between the β 1-strand and the rest of the protein is very unusual. Is this due to a crystallization artifact? The authors need to present all the protein complexes, if more than one, in one asymmetric unit as a supplementary figure.

Response:

Thank you for your constructive suggestions. We are sorry for the confusion about the structures. In fact, the SAP05 β 1-strand in the SAP05-GATA18 and SAP05-SPL5 structures cannot be fully observed in the electron density map, probably due to its flexible character. Instead, it can be clearly traced in the SAP05-RPN10 structure. In order to describe the overall structure of SAP05, we previously defined this region as a β 1-strand observed in the SAP05-RPN10 structure. Indeed, as suggested by the reviewer, we found that there are two copies of SAP05-RPN10 in one asymmetric unit, and the N-terminus of SAP05 packs against each other and forms an antiparallel β -sheet due to the crystal packing. To avoid confusion, we have redefined this as a loop and present the experimental structures in the revised figure (Fig. 1e, f and Supplementary Fig. 1b).

Fig. 1 | e, Ribbon diagram of the crystal structure of OY-M SAP05-AtGATA18 complex. Zinc ion is shown as a sphere. Four zinc-coordinating cysteine residues of AtGATA18 are numbered and shown in schematic form. **f**, Crystal structure of OY-M SAP05-AtSPL5 complex. AtSPL5 coordinates two non-interleaved Zn1 and Zn2 sites and is separated into an N-terminal subdomain (Nt-sub) and a C-terminal subdomain (Ct-sub).

Supplementary Fig. 1 | b, Crystal structure of the OY-M SAP05-AtRPN10 complex in one crystallographic asymmetric unit, where the N-terminus of SAP05 forms a β -strand owing to packing against its counterpart.

2. For reliable comparison of binding affinities, the ITC results for all the repeats need to be averaged with the standard deviation being provided, either in the figures or as a supplementary table. In addition, the binding stoichiometry (N value) needs to be provided. In Fig. 5i, the ITC binding assays for SAP05 with WT MqGATA18 needs to be presented for comparison.

Response:

Thank you for your constructive suggestions. For reliable comparison, we have provided all the ITC data with averaged K_D value and standard deviation which are summarized in Table S1, and the binding stoichiometry (N value) has also been shown. For a clearer comparison of the binding ability of SAP05 to mutant and wild-type MqGATA18, we co-presented the ITC assay results in the same panel as is shown in Fig. 5j.

Fig. 5 | j, ITC measurement of binding affinity of OY-M SAP05 to WT and mutant MqGATA18 (Y-R, M-K, N-R).

3. In Table 1, the $I/\sigma I$ for the highest-resolution shell of 8J48 and 8J49 entries is much higher than 2. On what considerations were the resolution cutoffs selected?

Response:

We are sorry for this blunder. When we initially process the crystal data, to guarantee the excellent “Completeness”, we inappropriately choose the “ $I/\sigma I$ ”. So we have re-processed the structural data with the highest-resolution shell of 8J48 and 8J49 entries is 1.65 and 2.30, respectively which possesses relatively high quality (Table 1). And these data have been re-submitted to PDB database.

4. There is a mis-incorporated number ‘1.5’ in Supplementary Fig. 6a.

Response:

We are sorry for this error, we have corrected this in the revised manuscript.

5. In discussion, the authors mentioned that they failed to identify why WBDLb SAP05 does not bind to SPLs. Yet no experimental data were presented. The authors may want to elaborate more details on their work in this regard.

Response:

Based on the sequence alignment of SAP05 homologs in divergent phytoplasmas, we found that the majority of residues required for SPL5 binding (marked by black triangles) are strictly conserved, except for Glu72 and Ala102 of WBDLb SAP05 (Supplementary Fig. 8a). However, swapping of these two residues with Ala and Glu residues present in the other homologs did not rescue the SPL5 binding ability (Supplementary Fig. 8b). Even multiple-site swapping of WBDLb SAP05 cannot successfully induce the SPL5 binding by our ITC assay. We have added these in the revised manuscript.

Supplementary Fig. 8 | Binding analysis of AtSPL5 with wild-type and mutant WBDLb SAP05. a, Sequence alignment of SAP05 homologs in divergent phytoplasmas. Identical residues are marked by lightblue background. The residues that directly interact with AtSPL5 are numbered and denoted by black triangles. The potential residues required for AtSPL5 binding are indicated by red frames. **b,** ITC measurements of binding affinities of AtSPL5 to wild-type and mutant WBDLb SAP05. The ITC titration of AtSPL5 to wild-type WBDLa SAP05 was presented here as control. NB, no apparent binding under our experimental conditions.

Response to Reviewer #2

Comments:

In this manuscript, Yan et al. investigate the molecular interactions of SAP05 with SPL and GATA transcription factors and RPN10. The work follows up on a previous study from another lab that describes the mechanism of how phytoplasma SAP05 effector mediates the degradation of SPLs and GATAs by recruiting the 26S proteasome receptor RPN10. Huang et al found that SAP05 does not interact with animal GATA and insect/human RPN10 homologs and showed that purified human 26S proteasomes can degrade plant SPLs in the presence of SAP05 and Arabidopsis RPN10. Yan et al built on this knowledge to generate crystal structures and identify residues that may play a role in the specificity of SAP05 binding to the plant proteins.

1. Except for providing details of molecular interactions, the study does not reveal new mechanistic insights. For example, the work does not reveal how the 26S proteasome is degrading SPL and GATA targets.

Response:

Thank you very much for your constructive suggestions. Exploring how the 26S proteasome degrades SPL and GATA targets will indeed improve the insight into the non-ubiquitin degradation mechanism by the 26S proteasome. However, to the best of our knowledge, 26S proteasome is a highly dynamic protein machine and even the

classical ubiquitin-mediated degradation mechanism is still under investigation, thus exploring the non-classical degradation mechanism of SPL and GATA target by 26S proteasome will be a complicated process that requires a lot of effort and time, which is beyond our scope so far. In this study, we mainly detailed the interaction of SAP05 in complex with SPL5, GATA18 and RPN10, and revealed the details of the molecular mechanism of SAP05 in recruiting SPL5 and GATA18 to RPN10. We believe that this to some extent lays the foundation for exploring the 26S proteasome-mediated degradation of SPL and GATA.

2. The study shows additional insights into which amino acids play a role in the interactions of SAP05 and RPN10. However, the work is incomplete. Firstly, the study lacks data that show that the SAP05 mutants are intact and remain correctly folded. To address this concern, it is necessary to demonstrate that the mutant proteins retain other activities and that their overall stability remains unaffected in the assays.

Response:

Thank you for your suggestions. To certify whether SAP05 mutants are intact and remain correctly folded in our assay, we carried out a GST pull-down assay by using GST-RPN10 to pull down the SAP05 mutant that defective in TFs binding in our ITC assay, the results indicated that all the mutants of SAP05 like the wild type can apparently pull down the RPN10 (Fig. R1a), indicating that these SAP05 mutants retain RPN10-binding activity albeit loss of TF-binding activity. To further confirm the overall stability of these mutants, we performed gel-filtration chromatography assay and showed that all SAP05 mutants exhibit a uniform non-oligomerization peak identical to wild-type (Fig. R1b), suggesting that SAP05 mutants are stable and correctly folded.

Fig. R1 | Protein stability analysis of SAP05 mutants. a, GST pull-down assay using GST-tagged RPN10 to pull down wild-type and mutant SAP05. **b**, Superdex 75 Increase 10/300 gel-filtration chromatography profiles of wild-type and mutant SAP05.

3. Data described in lanes 215 to 220 of the results misrepresent the data shown. This section claims that SAP05 mutants are studied, but the figure shows data for SPL5 mutants.

Response:

Thank you for pointing this. We have corrected this in the revised manuscript.

4. The majority of the 26S proteasome experiments included in the paper don't show evidence of degradation, and the few that may do, there is only marginal differences between the lanes, and moreover, the protein levels in the other lanes in the same experiments appear to go down as well. The 26S proteasome experiments include limited information on turnover rates and efficiency of degradation in relation to the concentrations of substrates provided.

Response:

Thank you for your great comment. We have repeated the 26S proteasome experiments three times. To more clearly exhibiting the relative degradation trends of the substrate, we present quantitative plot to compare the degradation effects over time (Fig. 4f, Fig. 5g, Fig. 5m and Supplementary Fig. 5l). The results is consistent with the description in our manuscript and confirm our conclusion.

Just as the reviewer mentioned, we found that there was occasional degradation of vWA and SAP05. In addition, we also found that the addition of MG132 (a proteasome inhibitor) did not result in the degradation of either SPL5 or vWA/SAP05, thus it is reasonable that during the TFs translocation into the proteasome CP chamber for degradation, partial SAP05 and/or vWA components are be dragged down along with TFs into the CP channel for degradation, but this conjoint degradation is minimal and does not affect our conclusion.

As per your suggestion, we have detailed the substrate concentration, incubation time and other related experimental information for the degradation assay and updated them accordingly in the revised manuscript.

Fig. 4 | f, Quantification of the percentage of retained GST-SPL5 in the degradation assay, corresponding to Fig. 4, a-e (Mean±S.E.M.; n = 3 independent experiments).

Fig. 5 | g, Quantification of the percentage of retained GST-GATA18 in the degradation assay, corresponding to Fig. 5e, f (Mean±S.E.M.; n = 3 independent experiments).

Fig. 5 | m, Quantification of the percentage of retained GST-GATA18 in the degradation assay, corresponding to Fig. 5k, l (Mean±S.E.M.; n = 3 independent experiments).

Supplementary Fig. 5 | l, Quantification of the percentage of retained GST-SPL5 in the

degradation assay, corresponding to Fig. 4a and Supplementary Fig. 5, i-k (Mean \pm S.E.M.; n = 3 independent experiments).

Response to Reviewer #3

Comments:

The present work approaches the mechanism of SP05-mediated degradation by the proteasome of plant transcription factors SPL5 and GATA18 by means of crystal structure and structure/function studies.

SAP05 is a very new player in research literature, with only three articles in PUBMED, starting in 2021. The main reference on SAP05 is an excellent work published in Cell (Huang W, MacLean AM, Sugio A, Maqbool A, Busscher M, Cho ST, Kamoun S, Kuo CH, Immink RGH, Hogenhout SA. Parasitic modulation of host development by ubiquitin-independent protein degradation. Cell. 2021 Sep 30;184(20):5201-5214.e12. doi: 10.1016/j.cell.2021.08.029. Epub 2021 Sep 17. PMID: 34536345; PMCID: PMC8525514.). Hogenhout group's work describes a very relevant and unprecedented mechanism of degradation of transcription factor by SAP05, hijacking one of the main receptors of the 26S proteasome, Rpn10, for directing substrates to the proteasome in an ubiquitination-independent fashion. Without going deeper on, it is clear that the mechanism uncovered by Hogenhout's paper is not only important in parasite biology, but also in proteasome, protein degradation and proteostasis fields, since well-detailed ubiquitin-independent protein degradation processes by the 26S proteasome are scarce in the literature.

Said that, the current paper becomes highly relevant since it complements and characterizes the mechanism proposed by Huang and collaborators. Furthermore, it describes with great detail the interactions between complex partners and the basis for the specificity of the mechanism. The experimental design and the methodology are outstanding and appropriate for the work. The results are remarkable, with several crystal structures solved, including subsequent interface definition, mutagenesis and functional assays. Furthermore, in general terms, it is correctly written and references are fine.

My suggestion is that the paper should be accepted with very minor revisions. However, there are aspects that should be addressed:

1. It is mentioned that, SAP05 “acts”, “behaves” or “functions” as a “PROTAC” that brings together individual TFs and AtRPN10 by means of two distinct lobes (Lines 82, 147 and 363). This analogy generates problems. Protacs are usually small-molecule based chimeras chemically synthesized that recognize specific motifs in targets, and they recruit E3-ligases to induce ubiquitination. SAP05 would act more as an adaptor with some analogies with the mechanism of recruitment and degradation of ornithine decarboxylase (ODC), or with the interaction FAT10 - Rpn10-vWA (even though the involved surfaces are not identical). The “PROTAC” concept could be maintained in the text, but in different terms. For example, by changing “behaves like a PROTAC” by “shows some similarities with proximity-based proteolytic chimeras” or similar

sentences. Moreover, it should be highlighted in the text that it is not explained how SAP05 avoids degradation. In the case of small Protacs it is evident that they are not proteins so they are not proteolyzed by the 26S, but SAP05 is a protein that binds Rpn10. Why it is not degraded?

Furthermore, ODC and FAT10 should be mentioned somewhere in the paper.

Response:

We agree with the reviewer. We have revised the statement related to “PROTAC” as suggested by reviewer.

We speculated two possible mechanisms to explain how SAP05 avoids degradation: During the initiation of TFs translocation into the proteasome CP chamber for degradation, SAP05 is expected to uncouple with TFs but remain with RPN10 due to the latter has a stronger bond with SAP05. Alternatively, in the processive translocation phase, the TFs undergo unfolding by the heterohexameric ATPase ring, and the unfolded TFs may lose their ability to bind to SAP05. Consequently, SAP05 may be released from TFs to spare them from degradation. Nevertheless, a small amount of SAP05 and/or vWA components are still co-transported with TFs into the CP channel for degradation, since a slight degradation of SAP05 and vWA was observed in our western blot assay. We have included this statement in the revised manuscript.

We have described the ODC and FAT10 reports in the Introduction section and cited the papers in the revised manuscript.

2. Line 108: “SPL5-AtRpn10” should be “SAP05-AtRpn10”

Response:

We are sorry for this carelessness, we have corrected this in the revised manuscript.

3. Line 256: “In eukaryotic cells, the UPS is responsible for the degradation of more than 80% of intracellular proteins” should have a reference.

Response:

Thank you for your suggestion, we have added the relevant references in our revised manuscript.

4. Proteasome purification and degradation assays: the way they are designed, human 26S proteasome are purified and vWA-Rpn10 from *Arabidopsis thaliana* is used to reconstitute the degradation (by ternary interactions with SAP05 and substrates). This is acceptable, and apparently it works, however it is not mentioned if purified 26S proteasomes are occupied by human Rpn10. It is important to check if human Rpn10 is stoichiometric, sub-stoichiometric or absent, in proteasomes purified for the assay. In methods section, the washing conditions are defined as: “streptavidin beads were washed 5 times with buffer A” (containing 100 mM NaCl). It is hard to know if endogenous Rpn10 will be retained or dissociated from the 26S, therefore, this information should be specified.

Response:

We thank the reviewer for the great suggestion. By Western blot analysis, we found that the human RPN10 can be detected on our purified 26S proteasomes by using anti-RPN10 antibody. Thus, the purified 26S proteasomes, such as our method, at least partially incorporate the endogenous RPN10. Next, the vWA competition assay showed that the addition of purified vWA can successfully compete with endogenous RPN10 for binding to the 26S proteasomes, as the supernatant RPN10 gradually increased with the introduction of additional vWA proteins (Supplementary Fig. 5a). In fact, the amount of vWA in our experiment is approximately 400-fold higher than that of purified 26S proteasomes, and thus sufficient to replace endogenous RPN10 from the purified 26S proteasomes. We have included this in the revised manuscript.

Supplementary Fig. 5 | a, Western blot analysis of vWA competition assay. As the molar ratio of vWA to RPN10 increased (0:1, 2:1, 4:1), more RPN10 was competitively replaced by vWA on the purified human 26S proteasomes.

5. Figure 4: it is too crowded and repetitive. I would recommend to limit it to 3-4 panels and the rest in the supplement.

Response:

Thank you for your suggestions, we have simplified the panel arrangements of Fig.4 in our revised manuscript, and the rest of relative results have been moved to Supplementary Fig. 5.

6. All Figures with ITC graphs: “moler ratio” should be “molar ratio”

Response:

We are sorry for this typo, we have corrected this in the revised manuscript.

REVIEWERS' COMMENTS

Reviewer #2 (Remarks to the Author):

The responses to reviewer 2, comments 3 and 4 are satisfactory.

The extra data in response to reviewer 2, comment 2 is appreciated, though it appears that these new data were not included in the manuscript itself. The experiment is a GST pull down and SAP05 and SAP05 mutants are pulled down in all experiments that have GST-RPN10. The last lane does not contain GST-RPN10, and shows that wt SAP05 does not pull down with GST alone. However, the experiment is missing essential controls to confirm the specificity of RPN10 interaction with SAP05 and to determine the optimal stringency of the experimental conditions. To control for these, it will be necessary to do a pull down of GST-RPN10 together with a SAP05 mutant that does not bind Rpn10 and with another protein, such as perhaps GFP.

I agree with the response to reviewer 2, comment 1 that conducting additional work on the mechanisms of how the 26S proteasome mediates degradation of SPLs and GATAs is beyond the scope of the manuscript. However, the abstract and introduction nonetheless promise mechanistic insights (for example, lines 28-29, lines 73-75). This may need to be rephrased to describe the presented work more accurately.

Additional minor comments:

Lines 59-61: To distinguish these examples from the processes involving SAP05, it would be informative to add that ODC requires modulation via interaction with antizyme (AZ) to be recognized by the proteasome. Furthermore, FAT10/NUB1L are ligands added to a substrate in a process reminiscent of the ubiquitin pathway in that it also requires E1/E2/E3 ligases and the presence of lysine residues on the substrate targeted for degradation. Whereas FAT10/NUB1L bind the VWA domain of Rpn10, they are degraded along with their substrates.

Line 36: Suggestion to change '... for the manipulation ...' to '... into the modulation ...'.

Additional minor comments:

1. Lines 59-61: To distinguish these examples from the processes involving SAP05, it would be informative to add that ODC requires modulation via interaction with antizyme (AZ) to be recognized by the proteasome. Furthermore, FAT10/NUB1L are ligands added to a substrate in a process reminiscent of the ubiquitin pathway in that it also requires E1/E2/E3 ligases and the presence of lysine residues on the substrate targeted for degradation. Whereas FAT10/NUB1L bind the VWA domain of Rpn10, they are degraded along with their substrates.

Response:

Thank you for your suggestions. We have added the relevant information to the Introduction section as suggested by the reviewer.

2. Line 36: Suggestion to change ‘... for the manipulation ...’ to ‘... into the modulation ...’.

Response:

Thank you for pointing this. We have corrected this in the revised manuscript.